# Endothelial Dysfunction: Is There a Hyperglycemia-Induced Imbalance of NOX and NOS?

**DOI:** 10.3390/ijms20153775

**Published:** 2019-08-02

**Authors:** Cesar A. Meza, Justin D. La Favor, Do-Houn Kim, Robert C. Hickner

**Affiliations:** 1Department of Nutrition, Food & Exercise Sciences, Florida State University, Tallahassee, FL 32306, USA; 2Institute of Sports Sciences and Medicine, College of Human Sciences, Florida State University, Tallahassee, FL 32306, USA; 3Department of Biokinetics, Exercise and Leisure Sciences, School of Health Sciences, University of KwaZulu-Natal, Westville 4041, South Africa

**Keywords:** endothelium, eNOS, glucose, hyperglycemia, insulin resistance, NADPH oxidase, obesity, reactive oxygen species, ROS, type 2 diabetes, vascular function

## Abstract

NADPH oxidases (NOX) are enzyme complexes that have received much attention as key molecules in the development of vascular dysfunction. NOX have the primary function of generating reactive oxygen species (ROS), and are considered the main source of ROS production in endothelial cells. The endothelium is a thin monolayer that lines the inner surface of blood vessels, acting as a secretory organ to maintain homeostasis of blood flow. The enzymatic production of nitric oxide (NO) by endothelial NO synthase (eNOS) is critical in mediating endothelial function, and oxidative stress can cause dysregulation of eNOS and endothelial dysfunction. Insulin is a stimulus for increases in blood flow and endothelium-dependent vasodilation. However, cardiovascular disease and type 2 diabetes are characterized by poor control of the endothelial cell redox environment, with a shift toward overproduction of ROS by NOX. Studies in models of type 2 diabetes demonstrate that aberrant NOX activation contributes to uncoupling of eNOS and endothelial dysfunction. It is well-established that endothelial dysfunction precedes the onset of cardiovascular disease, therefore NOX are important molecular links between type 2 diabetes and vascular complications. The aim of the current review is to describe the normal, healthy physiological mechanisms involved in endothelial function, and highlight the central role of NOX in mediating endothelial dysfunction when glucose homeostasis is impaired.

## 1. Introduction

Type 2 diabetes (T2D) is a metabolic disease, characterized by hyperglycemia, which often coincides with macro- and micro-vascular complications. Early onset of T2D increases the risk of cardiovascular disease (CVD), and the age at which patients are diagnosed with T2D is declining [1]. The prolonged exposure and progressive nature of cardiometabolic impairments in T2D likely underpins the alarming rate (70%) at which diabetic adults ≥65 years die from CVD [2]. Despite growing efforts to describe the molecular interactions between T2D and CVD, there are several common pathophysiological features that complicate treatment such as insulin resistance, chronic inflammation, oxidative stress and hypertension; therefore, it is important to understand the features of each condition to delay or prevent the onset of cardiometabolic disease.

The pathogenesis of T2D is not completely understood, however, there are two key precursors that lead to development of this disease: (1) peripheral insulin resistance and (2) β-cell dysfunction [3]. In a normal healthy state, glucose homeostasis is maintained by a reciprocal balance between β-cells’ response to rises in blood glucose and the subsequent glucose-lowering actions of insulin in peripheral tissues such as skeletal muscle, liver and adipose tissue [4]. Skeletal muscle is the primary site of insulin-stimulated glucose disposal and the liver is responsible for suppressing glucose production. A prevailing hypothesis is that excess nutrient consumption results in an inability for adipose tissue to meet the energy surplus; in turn, lipids spill over to ectopic sites and disrupt insulin signaling [5]. While this represents the metabolic and cellular derangements that occur in obesity-related insulin resistance, acute spikes in glucose are sufficient to impair metabolic function. That is, acute hyperglycemia has been shown to attenuate endothelium-dependent vasodilation [6], insulin secretion [7] in addition to promoting the release of inflammatory proteins [8]. Therefore, consideration of the multi-tissue regulation involved in controlling insulin sensitivity is required to understand how endothelial dysfunction relates to T2D. In addition, the contribution of hyperglycemia alone poses a risk for development of endothelial dysfunction and insulin resistance. Understanding the mechanisms that couple endothelial function with metabolic function can improve treatment of the various vascular complications associated with T2D. Given that T2D causes damage to blood vessels throughout the body, patients are at a greater risk for development of other chronic conditions such as diabetic kidney disease and diabetic retinopathy [9]. Notably, aberrations in endothelial function often precede many of the abnormalities observed in T2D and CVD [10].

The vascular endothelium is an active endocrine organ involved in the regulation of vascular tone and maintenance of vascular homeostasis. This monolayer of tissue lines the inner wall of blood vessels, hence positioning endothelial cells in direct contact with the flowing blood and resultant shear stress. Endothelial cell exposure to shear stress leads to transmission of mechanical signals to tightly control lumen size [11]. In addition, the delivery of circulating molecules such as insulin and glucose to skeletal muscle fibers is regulated by endothelial cells. Skeletal muscle accounts for approximately 80% of insulin-stimulated glucose disposal [12], and the hemodynamic forces due to shear stress augment insulin’s vasodilatory actions [13]; therefore, the endothelium is a critical factor in mediating the links between vascular function and metabolic demands. On the other hand, damage to the endothelium is characterized by phenotypic changes, inflammation, altered permeability as well as reduced endothelium-dependent dilation. Thus, the investigation of endothelial dysfunction has received widespread attention as a predictor of future cardiovascular events and insulin resistance. Evidence in animal models and cultured cells have identified a key role of oxidative stress during the development of T2D vascular complications [14,15,16,17]. During the shift from normal vascular physiology to endothelial dysfunction accompanied by hyperglycemia, there is an inability to balance the cellular redox environment, favoring excess production of reactive oxygen species (ROS) without the appropriate scavenging by antioxidants [18,19,20].

NADPH oxidases (NOX) are membrane-bound enzyme complexes that are considered potent stimulators of ROS in endothelial cells, particularly under conditions of hyperglycemia [21,22]. In physiological concentrations, NOX-derived ROS act as signal transducers, mediating biological processes such as endothelial cell angiogenesis and migration via vascular endothelial growth factor (VEGF) and the small G-protein, Rac1. The predominant NOX isoforms present in endothelial cells are NOX1, 2, 4, and 5 [23]. In addition, NOX are present in skeletal muscle and are major producers of ROS in this tissue [24], thus NOX are also involved in mediating control of glucose uptake. Skeletal muscle possesses abundant NOX2 and NOX4 with minimal involvement from NOX1 [25]. While the molecular targets vary among the NOX isoforms, each is implicated in the development of vascular dysfunction when excessively stimulated. La Favor et al. [26] recently identified that obese humans display elevated protein expression of NOX subunits gp91^phox^, p22^phox^, p47^phox^, and p67^phox^ in skeletal muscle.

This review will present the normal healthy mechanisms involved in generation of nitric oxide (NO) and highlight the recent progress in understanding the contributions of hyperglycemia-induced NOX activation in endothelial dysfunction, with a focus on the interplay between NOX and eNOS regulation. In addition, we will describe the interactions between endothelial NOX and maintenance of glucose homeostasis.

## 2. Normal Endothelial Function

A healthy endothelium modulates homeostasis of the vasculature in response to physical and chemical stimuli by secreting vasoactive molecules that act in autocrine, paracrine, or endocrine fashions. These endothelial-derived factors control a number of physiological outcomes including vessel tone, diameter, vascular smooth muscle cell (VSMC) proliferation, platelet activation, and leukocyte adhesion. In turn, the endothelium is considered a strong indicator of cardiovascular health, and several non-invasive techniques have been developed to assess endothelial function [27]. The importance of normal endothelial function can be highlighted by an integrative nature in establishing a direct link between vascular and metabolic outcomes. Elucidation of the healthy physiological mechanisms therefore provides a foundation for clinical strategies to treat cardiometabolic diseases.

Of the wide array of molecules generated by endothelial cells, nitric oxide (NO) may be considered central to mediating the diverse actions carried out by the endothelium. The seminal studies by Furchgott and Zawadzki introduced the pivotal role of NO in regulating endothelium-dependent vessel dilation [28]. Whereas acetylcholine (ACh) promoted vasodilation of isolated rabbit aorta in the presence of an intact endothelium, removing the endothelium led to Ach-induced vasoconstriction. The paradoxical effects of ACh were later designated to the presence or absence of NO in the endothelium, initially termed endothelium-derived relaxing factor. It is now understood that endothelial NO is mainly produced from L-arginine via the constitutively active, calcium-calmodulin (CaM)-dependent enzyme endothelial nitric oxide synthase (eNOS) to yield NO and L-citrulline. The eNOS molecule is comprised of a dimer with a zinc thiolate (ZnS_4_) core [29], and is activated in the presence of molecular oxygen (O_2_) and the cofactors heme, tetrahydrobiopterin (BH_4_), flavin adenine mononucleotide (FMN), flavin adenine dinucleotide (FAD), and NADPH [30]. The subsequently produced NO, a soluble gas, diffuses through the endothelium to target soluble guanylyl cyclase (sGC) in VSMC. In turn, sGC leads to increased guanosine diphosphate-guanosine triphosphate (GDP-GTP) exchange, cyclic guanosine monophosphate (cGMP)-mediated activation of protein kinase G (PKG) and reduced calcium concentrations [31], thereby promoting vessel relaxation (Figure 1). The β-subunit of the sGC protein is considered a physiological NO sensor [32] and has become a desirable pharmacological target as a hypertensive treatment to circumvent the complex mechanisms of NO production.

It is important to note that while eNOS is a critical factor in maintaining vascular function, eNOS uncoupling is associated with pathological conditions. That is, eNOS is “uncoupled” from the production of NO and instead generates superoxide (Figure 1). In turn, superoxide rapidly combines with NO to form the reactive nitrogen species (RNS), peroxynitrite (ONOO^−^) [33], thus reducing the bioavailable NO. Notably, the reaction between NO and superoxide occurs at a rate three times faster than the enzymatic reduction of superoxide by superoxide dismutase (SOD) [34], indicating that cytosolic superoxide concentrations have significant implications for endothelial dysfunction. ONOO^−^ is considered a highly reactive oxidant with the ability to cause several cellular derangements, including apoptosis and nitration of tyrosine residues [35]. Increased ONOO^−^ levels can subsequently generate a wide array of negative reactions associated with eNOS uncoupling, such as oxidation of BH_4_ [36], iron-sulfur centers [37], and the ZnS_4_ core of eNOS [38]. Oxidation of the ZnS_4_ core results in monomerization of eNOS and inhibition of NO production. As a result, the negative by-product of eNOS uncoupling, ONOO^−^, has the ability to exacerbate the dysfunctional NO production. Depletion of BH_4_ is considered a primary mechanism involved in eNOS uncoupling and is observed in the presence of oxidative stress [39]. Superoxide and ONOO^−^ can both catalyze the oxidation of BH_4_ to dihydrobiopterin (BH_2_), effectively limiting substrate availability for eNOS activation. The causative role of BH_4_ depletion in endothelial dysfunction can be demonstrated by restored endothelium-dependent dilation with overexpression of the BH_4_ rate-limiting enzyme, guanosine 5′-triphosphate cyclohydrolase 1 (GTPCH1) in apolipoprotein E knockout (ApoE KO) mice [40]. Additionally, BH_4_ treatment can partially restore eNOS dimerization in the presence of ONOO^−^, while oxidized BH_4_ causes a loss of eNOS activity [41]. These observations indicate that suboptimal levels of BH_4_ ultimately prevent NO production, while ONOO^−^ catalyzes the oxidation of BH_4_ and disrupts the eNOS dimer. Thus, the dysregulation of eNOS activation results in a harmful cycle of ROS production, where eNOS itself is considered a direct source of ROS. Given that NOX are a major source of superoxide in the endothelium, it can be expected that NOX contribute to eNOS uncoupling (Figure 1). This relationship between NOX and uncoupling of eNOS will therefore be addressed in Section 4 of this review. 

### 2.1. NOS Isoforms Mediate Endothelium-Dependent Dilation

Although eNOS is the most abundant and predominant source of NO in endothelial cells, there are other NOS isoforms expressed in endothelial cells and other cell types. The inducible NOS (iNOS) isoform is most prevalent in macrophages and is primarily ‘induced’ by an immunological stimulus [42]; therefore, increased iNOS expression in endothelial cells is often indicative of cytoxicity and pathology [43,44]. Notably, iNOS generates greater amounts of NO compared with eNOS, and gene transcription is the primary mechanism of iNOS activation [45]. Excess iNOS-derived NO can be deleterious, as the surplus of NO competes with eNOS for BH_4_ as a cofactor and promotes ONOO^−^ formation via interactions between NO and superoxide [39]. Genetic ablation of iNOS in ApoE KO mice reduces low-density lipoprotein (LDL) oxidation and advanced atherosclerotic lesions [46], suggesting iNOS is key in the development of atherosclerosis. Animal models of T2D have also indicated a deleterious role of iNOS related to insulin resistance [47,48,49], however skeletal muscle iNOS expression was reported to be similar between insulin sensitive non-obese compared with obese insulin resistance adults [50].

The neuronal NOS (nNOS) isoform, highly expressed in perivascular nerve fibers, has been considered atheroprotective in the endothelium, as nNOS may regulate vascular tone in the absence of eNOS [51]. The ability for nNOS to generate hydrogen peroxide (H_2_O_2_) allows the isoform to promote vessel dilation [52]. Moreover, there is accumulating evidence of alternative splice variants from the nNOS gene, *NOS1*, including nNOSμ, nNOSβ, nNOSγ, and nNOSα [53,54,55,56]. Initially, nNOSμ was implicated in the control of insulin sensitivity [57], although a recent report postulated that nNOSβ may be the primary nNOS variant responsible for the enhancements in glucose uptake following muscle contraction [55]. Non-selective inhibition of NOS via *N*^ω^-nitro-L-arginine methyl ester (L-NAME) prevents the muscle contraction-induced improvements in insulin sensitivity. However, the particular NOS isoform underlying these enhancements has remained elusive. There is a marked increase in glucose uptake from isolated skeletal muscles derived from both nNOSμ KO and wild type mice, which occurs with [55] and without [58] the presence of insulin. Hence, these results indicate that nNOSμ is not essential for mediating insulin sensitivity during exercise. However, the observations of attenuated insulin-stimulated glucose uptake following muscle contractions by NOS inhibition remain unexplained. Considering nNOSβ is the only other nNOS variant expressed in skeletal muscle [54], it is likely that exercise-induced improvements in insulin sensitivity involve nNOSβ.

The identification of different NOS isoforms in the context of endothelial function provides novel avenues for potential treatments against CVD; however, the interactions between the various types of NOS enzymes is not fully understood. Important considerations of NOS KO models include the compensatory actions exerted by other NOS isoforms as well as NOS-independent vasodilation. For instance, eNOS KO mice display enhanced vasodilatory sensitivity to sodium nitroprusside (SNP) due to inhibition of phosphodiesterases [59]. Thus, the observations of preserved dilatory responses with ablation of a particular NOS isoform may not be directly attributed to compensation from alternate NOS isoforms. In addition, eNOS activation, the most widely studied form of NO production, can be regulated by various molecular interactions and further complicates the understanding of endothelial function.

### 2.2. Shear Stress Evokes eNOS Activation

There are various stimulators of eNOS activation, including vasoactive factors (e.g., ACh, bradykinin, adenosine, VEGF, and serotonin); however, the shear stress applied to the vessel wall by blood flow is a primary mechanism by which NO-dependent vasodilation occurs. The exposure of mechanical forces to the endothelium arising from flowing blood converts the mechanical stimulus into a biochemical cue and initiates a signaling cascade to increase NO activity. Determining the mechanisms of eNOS activation have been challenging to define due to differences in the degree of shear stress experienced by vascular beds between body regions in addition to differences between in vitro and in vivo models that are studied [60]. Nonetheless, the various mechanisms controlling eNOS activation were recently outlined, and include phosphorylation, glutathionylation, *S*-nitrosylation, and *N*-acetyl glycosylation [43].

Activation of eNOS occurs via a balance between mechanisms dependent and independent of calcium concentrations. Shear stress increases cytosolic calcium concentrations [61], but also leads to several signal transduction events [62]. The eNOS Thr-495 residue is responsive to calcium levels, and is occupied under basal conditions due to phosphorylation by protein kinase C (PKC), subsequently inhibiting calcium-CaM-dependent activation. Whereas, dephosphorylation (i.e., activation) occurs by protein phosphatase 1 (PP1) with a rise in calcium levels. A different phosphorylation site, Ser-1177 (in humans) or Ser-1179 (in bovine and rodents), promotes activation of eNOS via protein kinase A (PKA) or protein kinase B (Akt/PKB) in response to shear stress and insulin, respectively [43,63,64]. In addition, PKA-dependent activation of eNOS occurs through PI3K-dependent and -independent mechanisms [65]. Shear stress-mediated activation of Ser-1179 is prevented by PI3K inhibitors, but Ser-635 phosphorylation by PKA is unaffected by wortmannin or LY-294002. The Ser-635 site is believed to support eNOS activation under basal conditions, as mutation of the residue to alanine does not affect NO release [66]. In hyperglycemic conditions, however, eNOS activation is attenuated by increased superoxide and *N*-acteyl glycosylation of Ser-1177 [67]. Moreover, the cysteine (Cys) residues Cys-689, Cys-908, and Cys-382 can be subjected to *S*-glutathionylation and promote eNOS uncoupling under conditions of oxidative stress [68]. Given the crucial role of eNOS in the vasculature, assessment of the various posttranslational modifications involved in eNOS activation can provide a comprehensive understanding of endothelial function. 

The incorporation of structural and biochemical signals between blood flow and eNOS is an important means to investigate deficits in endothelial dysfunction. There are several sensors in the vascular wall responsible for detecting shear stress and promoting “mechanotransduction” including mechanosensing ion channels, integrins, and G-protein coupled receptors (GPCR) [69]. The cytoskeleton of the endothelium activates integrin mechanoreceptors, upon physical disruption by shear stress, to promote translocation of signaling proteins. Recently, a model of the molecular mechanisms involved in mechanotransduction was proposed, suggesting an integration between several mechanosensors and cytoskeleton proteins [69]. Briefly, the mechanical stress imposed by laminar blood flow activates a cell-cell junction protein, platelet endothelial cell adhesion molecule (PECAM-1), forming a complex with a cadherin and VEGF receptor at the plasma membrane. Subsequent phosphatidylinositol-3 kinase (PI3K)-mediated activation of integrins promotes activation of several small GTPases that target the membrane-bound eNOS molecule [69]. Caveolae, membrane-bound domains involved in endothelial endocytosis of nutrients and hormones, interact with and inhibit eNOS at the plasma membrane. Increased calcium concentrations, heat shock protein 90 and small GTPases promote the dissociation of caveolae with eNOS allowing interaction with calcium-CaM [69,70]. The hemodynamic forces associated with blood flow and the ensuing mechanotransduction is an emerging area of investigation. While the physiological responses are not well understood, it is known that disturbed flow, such as in atherosclerosis, is linked with the vascular remodeling observed in CVD.

### 2.3. Coupling of Insulin’s Vasodilatory and Metabolic Functions 

The potent vasodilatory actions of insulin are critical for regulation of blood flow. Insulin increases compliance of large conduit arteries, promotes dilation of resistance arterioles, increases capillary recruitment and maintains capillary permeability to support nutrient delivery [71,72]. Insulin’s effects require a balance of downstream signaling events to maintain vascular tone in the basal state, while the vasodilatory response to insulin becomes pronounced in the postprandial state. In this regard, the skeletal muscle microvasculature provides a link between insulin’s vascular and metabolic functions by increasing the surface area for tissue perfusion. Considering skeletal muscle is the primary tissue for insulin-stimulated glucose disposal, these actions clearly underlie a principle role of the endothelium in regulating glucose homeostasis. 

The ability for insulin to stimulate a hemodynamic response in various vascular segments has been well-documented [73,74,75]. In the larger conduit arteries, insulin increases compliance by dilating the vessel [76]. Similarly, insulin dilates resistance arterioles and evokes blood flow to tissues by a mechanism that is NO-dependent [71]. Vincent and colleagues [73] demonstrated that muscle microvascular recruitment in response to insulin occurs prior to changes in limb blood flow, underlining the importance of skeletal muscle in regulating postprandial glucose disposal. Other studies have highlighted the ability of insulin to couple vascular and metabolic outcomes by demonstrating parallel increases in insulin-stimulated blood flow and insulin stimulated glucose disposal in humans [77]. These data indicate that insulin’s effects on the vasculature are important determinants in the subsequent glucose delivery to peripheral tissues. Further, it is known that physical activity confers improvements in the vasodilatory actions of insulin [78] and that shear stress increases acutely with exercise [79]. Whether muscle contractions or shear stress are responsible for adaptations in insulin-stimulated blood flow was not clear until a recent investigation by Walsh et al. [13]. Cultured endothelial cells and isolated skeletal muscle arterioles were subjected to 1 h of high shear stress (20 dynes cm^−2^), and adults underwent 1 h of unilateral lower-limb heating to evoke increased blood flow. During recovery periods, cells and arterioles were treated with insulin and the healthy men and women underwent systemic insulin infusion. The exposure to shear stress significantly augmented insulin-stimulated vasodilation independent of muscle contractions, regardless of species. That is, isolating the hemodynamic response to shear stress without muscle contraction or exercise is sufficient to improve vascular insulin sensitivity. Furthermore, the ratio of phosphorylated eNOS/phosphorylated mitogen-activated protein kinase (MAPK) was greater in cultured cells exposed to shear stress compared with controls [13].

Insulin signaling in endothelial cells is characterized by two divergent branches that either promote vessel dilation or constriction. Stimulation of the insulin receptor can either lead to NO-mediated vasodilation or MAPK-mediated vasoconstriction. The vasodilatory actions of insulin occur via activation of PI3K, Akt, eNOS phosphorylation of Ser-1177 and NO production [80]. In contrast, the opposing vasocontricting actions are attributed to insulin’s activation of MAPK-dependent signaling via Sos, the small GTP-binding protein Ras, Raf, mitogen-activated protein kinase kinase (MEK) and extracellular signal-regulated kinase 1/2 (ERK1/2) [81,82]. Similarities in insulin signaling have been observed between HUVEC and other metabolic tissues (e.g., adipose tissue and skeletal muscle), including upregulation of glucose transporter-4 (GLUT4) [80,83,84]. Therefore, the ability to induce peripheral glucose disposal relies on insulin to promote capillary recruitment and overall blood flow. These characteristics of insulin in the endothelium suggest that a fine balance between divergent PI3K/eNOS and Ras/MAPK signaling is required for maintenance of glucose homeostasis. Interestingly, evidence in animals and humans indicates that the PI3K-dependent pathway is selectively attenuated in insulin resistance without any effect on Raf/MAPK insulin signaling [81,85]. Disruption of these events by preferentially activating Raf/MAPK can exacerbate existing insulin resistance. A common feature of insulin resistance is hyperinsulinemia, thus excess insulin and glucose would only enhance insulin’s vasoconstricting effects and potentially damage the vessel walls. It is controversial whether the microvasculature becomes more permeable/leaky in T2D [72], but excess stimulation of MAPK by glucose is linked to development of diabetic nephropathy and retinopathy [86]. 

Ferri et al. [87] demonstrated that insulin’s striking paradoxical capacity to activate vasoconstriction occurs via ET-1. HUVEC incubation in insulin led to a linear rise in ET-1 protein and remained elevated for 24 h. The release of ET-1 from HUVEC to cultured medium was inhibited by a tyrosine kinase inhibitor and protein synthesis inhibitor, suggesting that insulin induces protein synthesis of ET-1, rather than release from an intracellular store. Following the release from endothelial cells [88], ET-1 acts in a paracrine fashion to promote vessel constriction via one of two GPCRs (ET_A_ and ET_B_) on VSMC [89]. In humans, increased plasma ET-1 is observed across various metabolic populations upon exposure to an insulin bolus, including in healthy [90], obese hypertensive, and T2D normotensive males [87]. It should be noted, however, that ET-1 levels in adults with metabolic syndrome are higher in arterial blood compared with deep-venous blood, while the opposite relationship is observed in healthy controls [90]. Thus, ET-1 clearance is upregulated in a healthy state and insulin resistance is accompanied by greater ET-1 secretion. These data suggest that maintenance of vessel tone is balanced between ET-1-mediated constriction and NO-mediated dilation. Whereas, insulin resistance upregulates ET_A_ expression in obese Zucker rat aortas [91] and ET-1 is responsible for the augmented constricting tone and endothelial dysfunction in human obesity and T2D [92,93,94]. Blockade of the ET-1 receptor, on the other hand, attenuates the vasoconstriction response in humans [95] and reduces atherosclerotic lesions in ApoE KO mice [96]. 

The observations of divergent insulin signaling pathways yield important insights into the mechanisms that underlie aberrant endothelial function and the subsequent metabolic impairments. The counterbalance between eNOS and MAPK signaling is consequently being investigated to determine links between CVD and T2D, with ET-1 acting as an important molecule regulating the development of both diseases (Figure 2).

## 3. Mechanisms of NADPH Oxidase Function

NOX are ROS-producing enzyme complexes, originally identified in phagocytes [97,98], and since discovered in endothelial cells [99]. The NOX subunits gp91^phox^, p22^phox^, p67^phox^, and p47^phox^ were first identified in cultured HUVEC by Jones and colleagues [99], confirming the system as a non-phagocytic source of oxidant production. The subsequent identification of the NOX homologues by other groups provided a framework to define the mechanisms of the redox environment in blood vessels. Remarkably, NOX have the primary function of generating ROS in a normal healthy physiological state in addition to being susceptible to stimulation in pathological conditions.

The four NOX isoforms expressed in the vasculature (NOX1, 2, 4, 5) differ in their subcellular localization, yet all function via electron transfer from cytosolic NADPH to molecular O_2_, thereby producing superoxide or H_2_O_2_. Each NOX isoform is distinguished by its core transmembrane catalytic subunit that involves integration with cytosolic subunits for full activation, with the exception of NOX4 and NOX5, which do not require cytosolic proteins for activation. The prototypical phagocytic NOX complex contains the following subunits: a NOX core catalytic subunit (i.e., NOX1, NOX2, etc.), p22^phox^ transmembrane stabilizer, p47^phox^ organizer, p67^phox^ activator, p40^phox^ organizer, and the small GTPase Rac1 involved in tethering of p67^phox^ to the membrane (Figure 3). Progress in characterizing NOX in other cell types has demonstrated a high degree of conservation among structural components. Recent elucidation of the NOX crystal structure revealed core NADPH- and FAD-binding domains, six transmembrane domains, and two transmembrane heme groups [100]. These findings demonstrated an ability of the NOX core to fine-tune its conformation within the cytoplasmic carboxyl (COOH) domain, altering the affinity for NADPH binding. This structural modification likely underscores the ability for NOX to regulate the cellular oxidative environment by controlling the amounts of NADPH-binding and subsequently the level of ROS production.

The assembly of the NOX core with its cytosolic subunits (i.e., activation) has been largely characterized in the NOX2 isoform, demonstrating a coordination of several protein-protein interactions. Upon stimulation, Rac induces GDP-GTP exchange and translocates to the membrane where it interacts with the core subunit. The organizer homologues, p47^phox^ and NOXO1, act as scaffolds by binding to phospholipids at the plasma membrane via phagocyte homology (phox) domains in the NH_2_-terminals. The Sarc homology domains (SH3) and proline-rich regions (PRR) of p47^phox^ allow interactions with p22^phox^ and p67^phox^, respectively. At this point, Rac can interact with the activator homologues, p67^phox^ and NOXA1, at the plasma membrane thereby completing formation of the NOX complex and increasing ROS production. Upon activation, NADPH is recruited to act as an electron donor for oxygen. Cytosolic NADPH first transfers electrons to FAD following p67^phox^ activation. The binding of NADPH to hemes found in the core subunit then facilitates electron transfer from the reduced FADH_2_ to a cytosolic heme. A heme in the outer membrane receives a second electron from an inner heme and finally donates to oxygen localized in the extracellular space. Oxygen is the electron acceptor to generate superoxide in most NOX isoforms; however, NOX4 displays a preference for H_2_O_2_ formation. 

The distinct biochemical properties of superoxide and hydrogen peroxide can lead to divergent physiological outcomes [101]. Superoxide has a negative charge that imparts high reactivity and creates a short-lived, unstable molecule. An important characteristic of superoxide is that the anion cannot traverse across plasma membranes. Increased superoxide production can consequently have deleterious actions in the endothelium, such as NO quenching to generate ONOO^−^ and inactivation of mitochondrial proteins [34]. In addition to the rate of production, superoxide concentrations are dependent on the levels of superoxide dismutases (SOD) and the non-enzymatic conversion to H_2_O_2_ [101]. The rapid conversion of superoxide to H_2_O_2_ can provide vascular protection given the higher stability of H_2_O_2_, although the high affinity between superoxide and ONOO^−^ can limit the actions of SOD under conditions of oxidative stress. The ability of H_2_O_2_ to modulate signal transduction and gene expression relies, in part, by the ability to cross plasma membranes via diffusion and aquaporins [102]. However, it should be noted that H_2_O_2_ has also been considered deleterious in high concentrations and can lead to BH_4_ depletion [103,104]. Furthermore, there are differences between superoxide and H_2_O_2_ in the antioxidants that counteract the two oxidant species. The main antioxidants of the vasculature are SOD, catalase, and glutathione peroxidase (GPx) [105]. While SOD is the primary antioxidant that catalyzes the reduction of superoxide, GPx and catalase are largely responsible for converting H_2_O_2_ into water and O_2_. The GPx reaction is facilitated by reduced glutathione (GSH), which provides an electron, to reduce H_2_O_2_. In turn, the ratio of GSH to the oxidized glutathione disulfide (GSH/GSSG) is an indicator of the cellular oxidative environment [106]. These distinct properties between superoxide and H_2_O_2_ are believed to partially explain the different functional outcomes observed between NOX4 and other NOX isoforms.

The basis for H_2_O_2_ production versus superoxide by NOX4 is incompletely understood. Several reports have indicated that H_2_O_2_ is the main oxidant species produced by NOX4 [107,108,109,110] while others have argued that NOX4-mediated superoxide production is not detectable due to membrane compartmentalization of NOX4. The cell surface localization of many NOX isoforms allows detection of superoxide production via most assays, however NOX4 has been identified along internal membranes of the endoplasmic reticulum [111], nucleus [112], and mitochondria [113]. Furthermore, the superoxide anion is unable to pass through lipid membranes, while the neutral H_2_O_2_ molecule is readily diffusible. Thus, it can be postulated that the NOX4-mediated superoxide is rapidly dismutated within membrane compartments, rendering it elusive to most detection methods involving an extracellular probe. Nisimoto et al. [110] demonstrated that both superoxide and H_2_O_2_ are produced by NOX4 in a detergent-solubilized, partially purified preparation of NOX4 that eliminated membrane compartmentalization. However, H_2_O_2_ accounted for 80% of the ROS production, which suggests that NOX4-mediated H_2_O_2_ production is an intrinsic property of the isoform. The distinct generation of H_2_O_2_ from NOX4 has been attributed to an extra-cytosolic loop (E-loop) that when ablated in vitro, switches NOX4 from mainly producing H_2_O_2_ to superoxide [108]. Others demonstrated that H_2_O_2_ is the main ROS product derived from NOX4 regardless of intracellular location [109], and transgenic overexpression of NOX4 in mouse endothelium leads to significant elevations of H_2_O_2_ without altering superoxide levels detected via a high-performance liquid chromatography (HPLC)-based method or the expression of SOD1, 2, and 3 isoforms [114]. The accumulated evidence indicates H_2_O_2_ as the main oxidant species produced by NOX4, while the mechanisms responsible for this observation are controversial. Nevertheless, targeting NOX4 in vascular pathology remains of interest due to the potentially beneficial features of NOX4-mediated H_2_O_2_ production.

In healthy conditions where the appropriate antioxidant defense is present, NOX mediate a balanced cellular redox environment that is optimal for metabolism. Pathological states, as in T2D and hypertension, conversely present overactive NOX that cause several physiological abnormalities [115]. For example, excess NOX-derived ROS activates transcription of pro-inflammatory genes [116] and attenuates antioxidant enzymes [117]. In certain conditions, NOX may increase ROS production from other subcellular compartments such as the mitochondria. As a consequence, oxidative stress can be perpetuated by NOX. For these reasons, NOX represent promising targets for treatment and prevention of vascular complications. It is also noteworthy that NOX4 is often considered protective rather than detrimental to endothelial function. Recent advances in knowledge of the structure and function of NOX, as described above, provide molecular explanations for the roles of these enzymes in vascular function. The following sections will describe how aberrant NOX activity is involved in endothelial dysfunction, particularly when hyperglycemia is present. 

## 4. Endothelial NOX-derived ROS Production Increases with Exposure to Hyperglycemia

Endothelial dysfunction refers to the diminished ability of the endothelium to perform physiological actions of vasodilation, vascular permeability, and maintenance of vessel tone. Compromised endothelial function is observed in people with insulin resistance and metabolic syndrome, often preceding the onset of disease by several years. In T2D patients, the primary metabolic characteristic is chronically elevated blood glucose, although it is generally accepted that inflammation and oxidative stress are other prominent features [118]. The exposure of endothelial cells to hyperglycemia results in an array of negative effects that compromise vascular function; hyperglycemia is therefore considered a major candidate in the development of endothelial dysfunction in T2D. Accumulation of ROS, advanced glycation end products (AGEs), and altered cell signaling are just a few of the negative byproducts associated with hyperglycemic conditions. Among these endothelial cellular events triggered by high blood glucose, NOX are key players involved in mediating eNOS uncoupling and potentiating oxidative stress.

### 4.1. Endothelial Cell Culture Models of Hyperglycemia

There is evidence of excess NOX-derived ROS from studies using various in vitro models of hyperglycemia, although the relative roles of particular NOX isoforms remain to be determined. Studies in cultured endothelial cells have demonstrated increased ROS production primarily from NOX1 and NOX2. In human aortic endothelial cells (HAEC) cultured under normal (5.5 mmol/L) or high glucose (30 mmol/L) concentrations, protein expression of NOX2, NOX4, and p47^phox^ increased with high glucose [119]. In addition, NOX-derived ROS production increased concomitant with reduced eNOS expression. This attenuation of eNOS was abrogated by the non-selective NOX inhibitor, apocynin, which returned NOX2 and p47^phox^ to levels comparable with control and normal glucose conditions. In contrast, a study in mouse microvascular endothelial cells determined that hyperglycemia conferred greater p22^phox^ as well as greater eNOS expression; however, this was only observed after a longer duration of high glucose (72 h compared with 48 h) [120]. Results from a separate study are in agreement with increased eNOS expression associated with hyperglycemia. There are elevations in eNOS and NO in HUVEC exposed to high glucose (10, 25, and 50 mmol/L) for 4 and 8 h, in mice with STZ-induced diabetes, in adults with T2D, and in adults with T2D and coronary artery disease (CAD) [121]. NO production was not different between hypertensive adults and controls, however iNOS was also elevated in the HUVEC after hyperglycemic exposure. It can be postulated that insulin resistance in the two T2D cohorts attenuates the ability for insulin to induce endothelium-dependent vasodilation and instead there is a preference toward MAPK-dependent signal transduction. This would explain the significant elevations in iNOS expression and the compensatory production of NO only observed in populations with high glucose. Indeed, increased NO is observed in obese mice fed a high fat diet, whereas KO of iNOS reduces NO levels and protects against insulin resistance [122]. However, the findings related to NO production in the cells treated with hyperglycemia are less clear due to differences in concentrations and duration of exposure. In addition, the potential roles of oxidative stress and NOX-derived ROS were not reported in the study with varying durations of high glucose treatment [121]. 

The upregulation of NO may be considered a beneficial outcome to mitigate the potential imbalance in insulin signaling and vasoconstriction; however, greater NO also increases the risk of interaction with superoxide. Uncoupling of eNOS is present in T2D [123], although the relationship between activation of specific NOX isoforms and eNOS uncoupling in T2D is not clear. HUVEC exposed to high glucose concentrations display increased proteasome-dependent degradation of GTPCH, thereby reducing the available BH_4_ [124]. Elevations in ONOO^−^ further increase the proteasome activity, and a separate study demonstrated that the release of zinc from GTPCH by ONOO^−^ in diabetic mice and cultured bovine aortic endothelial cells leads to increases in ubiquitination of GTPCH and inhibition of GTPCH enzyme activity [125]. In HAEC and isolated aortas transfected with NOX5, the increased superoxide production led to greater eNOS activation and NO production [126]. Yet, the increased NO did not translate to improved endothelium-dependent dilation, suggesting a significant proportion of NO produced in response to NOX5 is biologically unavailable due to oxidative stress. Rather, dysfunctional eNOS may be upregulated in an attempt to compensate for deficits in NO levels. Given the susceptibility of BH_4_ and ZnS_4_ to oxidation, NOX are likely major sources of eNOS uncoupling in T2D. A different study found that angiotensin II (Ang II) leads to NOX4-mediated increases in ONOO^−^ in addition to uncoupling of eNOS in glomerular mesangial cells [127]. Thus, excess ROS production is associated with a reduced ability to produce NO, and the role of specific NOX in mediating eNOS uncoupling presents an interesting topic for future investigations. 

It has been reported that NOX2 is not the predominant NOX isoform activated upon exposure to hyperglycemia. In murine brain microvascular endothelial cells, high glucose concentrations led to a rise in total NOX activity, ROS levels and apoptosis [128]. In support of a link between hyperglycemia and NOX1, only mRNA and protein expression of NOX1, but not NOX2 or NOX4, increased with high glucose. Subsequent siRNA-mediated knockdown of NOX1, however, abolished the total NOX activity. The mechanism of this reduction was attributed to an inhibition of the nuclear transcription factor, nuclear factor kappa-light-chain-enhancer of activated B cells (NF-κB), when cells were treated with resveratrol [128]. Increased NF-κB activation has been previously described as a strong stimulus of inflammation and metabolic impairments in T2D [129], and this study extends the negative effects of hyperglycemia to the brain microvasculature due to NOX1 activation. In contrast, NOX4 was identified as the primary source of ROS generation following NF-κB activation in HAEC [130]. Hyperglycemia enhanced the interaction between NF-κB/p65 and the NOX4 promoter, underlying the significant elevations in NOX4 mRNA and protein expression as well as increased ROS levels compared with normal glucose concentrations. The upregulation of NOX4 expression with hyperglycemia was attenuated by pharmacological inhibition of NF-κB/p65 via caffeic acid phenethyl ester (CAPE) and siRNA against NF-κB/p65. Further, treatment with rosiglitazone prevented NF-κB/p65 phosphorylation, nuclear translocation, and interaction with the NOX4 promoter [130]. Hence, this study elucidated that the mechanism of hyperglycemia-associated NOX4 expression occurs via NF-κB/p65 transcription. It should be noted, however, that the differences in vascular beds, species, as well as magnitude and duration of hyperglycemic exposure may influence NOX activation, including the particular isoform that is targeted. Nonetheless, these studies suggest that hyperglycemia is a potent stimulus for NOX-mediated ROS production in endothelial cells. Importantly, the deleterious effects of hyperglycemia can be abolished by rosiglitazone and resveratrol.

The major sensor of cellular energy balance is considered 5’-AMP-activated protein kinase (AMPK) [131], therefore drugs that target AMPK are commonly used to treat T2D. AMPK is primarily activated in response to alterations in cellular AMP/ATP ratio, but there is increasing evidence demonstrating that AMPK is also responsive to ROS levels [132,133,134]. The polyphenol, resveratrol, and the class of anti-diabetic medications, thiazolidinediones (TZD) both target AMPK, albeit indirectly [135,136]; hence, the interplay between AMPK and NOX activation has been examined. In cultured endothelial cells treated with only modest hyperglycemia (10 mmol/L), cells displayed elevated ROS production including increased protein expression and translocation of the NOX subunits p47^phox^ and Rac-1 [137]. Rosiglitazone, a TZD, conversely led to AMPK activation which subsequently inhibited PKC and translocation of p47^phox^ and Rac-1 [137]. These data indicate that PKC-dependent phosphorylation of p47^phox^ is a mechanism required for NOX activation, and others have demonstrated that PKC is ROS-sensitive [138]. Indeed, PKC-ζ phosphorylates endothelial p47^phox^ in the presence of tumor necrosis factor-α (TNF-α) [139] and the PKC-β1 isoform promotes hyperglycemia-induced apoptosis of endothelial cells [140]. In addition, pharmacological inhibition of PKC via chelerythrine reduces endothelial superoxide production in isolated human diabetic vessels [141] and PKC-δ increases NOX-mediated ROS production in adipocytes of insulin resistant mice [142]. Similarly, metformin, the most commonly prescribed T2D medication [143] and a potent AMPK activator, has been shown to inhibit the PKC-β2 isoform and reduce ROS production [144]. These studies corroborate previous findings related to the mechanisms of NOX-derived ROS production under hyperglycemia; high glucose-mediated activation of NOX are attributed to diacylglycerol (DAG) and PKC signaling by promoting phosphorylation of the subunits p47^phox^, p67^phox^, and p22^phox^ [138,145]. Furthermore, evidence of AMPK’s protective effects related to NOX activation were described in mice with a deficient AMPK alpha 2 subunit (AMPKα2). The siRNA-mediated knockdown of AMPKα2 led to increased expression of NOX subunits as well as ROS production compared with wild types [146]. In addition, the transgenic mice displayed greater NF-κB activation including proteasomal degradation of its inhibitor, inhibitor of NF-κB-alpha (IκBα), and reduced endothelium-dependent dilation. Therefore, AMPK is a direct suppressor of NOX-derived ROS and possesses protective effects related to endothelial function. These findings provide mechanistic insight related to the improvements in vascular redox balance mediated by AMPK, however, a disconnect remains between the observations in cultured cells and the effectiveness of pharmacological activation of AMPK in humans. While metformin is widely recommended to treat T2D and reduces the risk of cardiovascular disease irrespective of glycemic control [147], the effectiveness of resveratrol and rosiglitazone in clinical studies is equivocal [148,149]. Future studies should determine the precise mechanisms by which AMPK targets NOX in hyperglycemia.

There is growing evidence that acute fluctuations in blood glucose are more detrimental to vascular health than constant high blood glucose [150]. As it relates to NOX activation, cells exposed to glucose fluctuations indeed generate greater levels of NOX-derived ROS [145]. Cells incubated in intermittent concentrations of normal and high glucose display significantly greater oxidative stress and protein expression of NOX subunits compared with cells exposed to constant high glucose. These events are both attenuated by PKC inhibition [145], suggesting similar mechanisms are involved regardless of the type of exposure, yet NOX activation is enhanced by acute spikes in glucose. Fluctuations in glucose also lead to uncoupling of eNOS in HUVEC via downregulation of GTPCH1 [151]. Metformin, conversely leads to the restoration of GTPCH1 and BH_4_ levels while attenuating ROS production and p47^phox^ expression [151]. Furthermore, the intermittent nature of glucose concentrations, as experienced following a meal, has also been reported to contribute to stress-induced senescence [152]. The elevations in p22^phox^ expression and superoxide production due to glucose fluctuations lead to a greater degree of endothelial cell senescence and DNA damage compared with constant high glucose [152]. Notably, three days of fluctuating glucose is sufficient to increase expression of p16 and p21 in HUVEC. Cellular senescence is a hallmark of the aging process [153], thus accelerated cell senescence associated with oscillations in blood glucose requires further attention. These findings have significant implications related to understanding how daily fluctuations in blood glucose contribute to chronic detriments in glucose homeostasis and the resultant vascular complications.

While NOX are major sources of endothelial ROS production and determinants of endothelial dysfunction, there is ample evidence in various endothelial cell lines indicating the mitochondrial electron transport chain (ETC) as the main source of hyperglycemia-induced ROS [154,155,156,157]. However, there is a possibility that cross-talk exists between NOX isoforms and other cellular compartments. Nevertheless, it has been proposed that the predominant source of ROS derived from endothelial exposure to hyperglycemia are NOX [158]. HUVEC incubated in 30 mmol/L of glucose display a significant rise in ROS production that is not abrogated by blocking mitochondrial ROS. Whereas the NOX inhibitors DPI and VAS2870 attenuate ROS levels, treatment with a mitochondrial uncoupler (CCCP) or inhibitor of complex I (rotenone) have no effect on ROS levels. The alternative hypothesis is the interaction between NOX and mitochondria, where ROS-induced ROS production can potentiate oxidative stress. This phenomenon was shown in cancer cells that exhibited supraphysiological ROS levels and resultant cell death due to an interplay between NOX and mitochondria [159]. Considering NOX4 is localized along the inner mitochondrial membrane [113], the potential interaction between these two sources of ROS may be significant determinants of the total ROS production in the endothelium. In this regard, it was demonstrated that hyperglycemia leads to colocalization of NOX4 and mitochondrial superoxide in isolated rat kidney and cultured mesangial cells [113]. This poses an intriguing challenge for future investigations to determine the relative production of mitochondrial ROS due to the electron transport chain and mitochondria-localized NOX4.

The influence of T2D on endothelial NOX5 is an emerging area of research; however, the NOX5 gene is not present in rodents, which creates a challenge for researchers to characterize the functions of this isoform in the endothelium. NOX5 has been observed throughout the human vasculature, including in the endothelium [160], VSMC [161], and cardiomyocytes [162]. Studies of NOX5 in cultured human cells have identified features distinct from NOX1, 2, and 4, as NOX5 is the only isoform that does not require intracellular subunits for activation and the N-terminal domain contains four calcium binding sites [21]. Another distinction of NOX5 includes the observation of six splice variants: NOX5-L (NOX5α, NOX5β, NOX5γ, NOX5δ), NOX5ε, and NOX5ζ [163]. While little is known about the specific function of each splice variant, NOX5α and NOX5β were reported to have the highest expression in isolated human VSMC and human microvascular endothelial cells [164]. In addition, treatment of the isolated cells with Ang II, ET-1, and TNF-α increased the expression levels of NOX5α and NOX5β. To circumvent the absence of NOX5 in rodents, accumulating studies use the approach of transgenic animal models expressing human NOX5 in a cell-specific manner [163,165]. A recent study transfected human NOX5 selectively in endothelial cells and VSMC of diabetic mice and demonstrated that diabetes leads to upregulation of both endothelial and VSMC NOX5 expression [166]. The mechanism of NOX5 activation was PKC-dependent, and results from an independent study demonstrated that PKCα is the isoform that mediates activation of NOX5 in HUVEC treated with high glucose concentrations [167]. Other studies have indicated that NOX5 mediates the expression of adhesion molecules, thus accelerating the atherosclerotic process [165]. Given that several pathological states display increased NOX5 activation, and that increasing studies are generating NOX5-expressing mice, it is expected that the molecular regulations of NOX5 will soon be described.

Collectively, the evidence of hyperglycemic exposure to cultured endothelial cells indicates a prominent role of NOX-derived ROS in perpetuating endothelial oxidative stress (Figure 4). Irrespective of the durations and concentrations of high glucose treatment, excess NOX activation yields a milieu of unfavorable outcomes including reduced NO bioavailability and endothelial cell senescence. Given that most studies have not directly compared between different NOX isoforms, it remains unclear how much of the total NOX activity due to hyperglycemia can be partitioned to each isoform. The observations of NOX subunit translocation with high glucose suggest the involvement of NOX1 or NOX2, yet the evidence of NF-κB-mediated NOX4 transcription cannot be disregarded. These cell culture studies, nonetheless, determine mechanisms of NOX activation and present opportunities to further the understanding of NOX in T2D and CVD. Importantly, T2D and CVD are complex, multi-tissue diseases therefore in vivo investigations are necessitated to enhance treatment strategies. 

### 4.2. Endothelial NOX Activity in Animal and Human Clinical Studies 

Much of what is known about NOX and the pathophysiology of vascular disease is due to studies in transgenic animal models and include models of atherosclerosis, hypertension as well as T2D. Interestingly, the early studies of NOX in experimental animal models focused on the etiology of atherosclerosis and hypertension, while most transgenic NOX models of obesity and T2D emerged within the last decade. Similar to the in vitro findings, animal models of insulin resistance demonstrate roles for different NOX isoforms. Importantly, NOX1 and NOX2 are consistently shown to be harmful to the vasculature [168,169,170,171,172,173,174,175,176,177], while defining the role of NOX4 has been elusive. Oxidative stress is a critical factor involved in several diabetic complications such as diabetic nephropathy and diabetic retinopathy, and the pathogenesis of these features has been reviewed elsewhere [22]. The following section, rather, will discuss how experimental manipulation of the various NOX isoforms in diabetic states contributes to impaired endothelial function. In addition, the regulation of glucose homeostasis by NOX will be discussed.

#### 4.2.1. NOX1

There is evidence of a detrimental role for NOX1 in animal models of dysregulated glucose homeostasis [168,169,170]. Wendt et al. [168] first demonstrated that eight weeks of STZ treatment led to a two-fold increase in rat aortic NOX1 protein, which occurred in parallel with reduced endothelium-dependent vasodilation. However, there were concurrent observations of increased eNOS expression and xanthine oxidase activity leaving a degree of uncertainty regarding the prevailing source of endothelial dysfunction. 

Induction of eNOS uncoupling comprises a significant source of ROS in diabetic animal models and impairs the ability to repair endothelial tissue damage [123], and NOX1 has been suggested as the predominant NOX isoform underlying the uncoupling of eNOS in diabetes [169]. STZ-induced diabetic mice exhibit a doubling of L-NAME-sensitive superoxide production [169]. However, the rise in superoxide and reduced endothelium-dependent vasodilation was abolished by p47^phox^ KO and siRNA-mediated knockdown of the NOX1 organizer subunit, NOXO1. In contrast, siRNA-mediated knockdown of NOX4 and inhibition of the ETC complex I by rotenone had no effect on superoxide production and endothelium-dependent dilation [169]. These findings suggest a selective contribution from NOX1 in uncoupling of eNOS and the subsequent impairment in endothelial function. However, the assessment eNOS uncoupling in this study was derived from the levels of superoxide following NOS inhibition via L-NAME. Although mitochondrial ROS did not contribute to endothelial dysfunction, and eNOS uncoupling is typically accompanied by excess superoxide generation, there are more robust methods of detecting eNOS uncoupling such as identifying the monomer/dimer ratio and assessing post-translational modifications [43]. Nonetheless, the specific role of NOX1 in mediating endothelial dysfunction was further corroborated by rejecting the notion of cross-talk between NOX1 and NOX2 in a p47^phox^-NOXO1 double-KO mouse model [170]. Given that NOXO1 and p47^phox^ are the organizer subunits for NOX1 and NOX2, respectively, and p47^phox^ also binds NOX1 [23], it was postulated that abolishing the action of one NOX subunit in diabetes would lead to compensatory upregulation of the counterpart. It was found that KO of either NOXO1 or p47^phox^ conferred the same antihypertensive effects and protection against endothelial dysfunction as the combined deletion of the two subunits, and there were no additive effects from the p47^phox^-NOXO1 double-KO [170]. However, NOXO1 KO alone presented a reduction in inflammatory genes such as interferon gamma (IFN-γ) while p47^phox^ KO displayed a pro-inflammatory signature. Therefore, NOX1 activation in diabetes may also be related to an immunological response. Indeed, diabetic ApoE KO mice demonstrate greater macrophage accumulation with NOX1-, but not NOX4- derived ROS [171,178]. These results indicate that the two organizer subunits, p47^phox^ and NOXO1, are responsive to the diabetic state and upregulate NOX1 activation by different mechanisms. As the complete deterioration of β-cell function occurs in advanced T2D [179], the data in animal models of dysregulated hyperglycemia suggest that chronic hyperglycemia may result in excess NOX1-derived superoxide and uncoupled eNOS, thereby attenuating NO production and endothelium-dependent dilation. 

Although it seems clear that NOX1 contributes to diabetes-associated endothelial dysfunction, it is difficult to make assertions regarding the role of NOX1 in the control of glucose homeostasis. To our knowledge, there are no studies that have investigated whether endothelial NOX1 has deleterious effects on peripheral glucose disposal. While NOX1 is not abundantly expressed in peripheral insulin sensitive tissues such as skeletal muscle [25], it is conceivable that the vasodilatory actions of insulin would influence endothelial NOX1. Among the three studies of diabetic animal models described above, the study by Youn et al. [169] reported that p47^phox^ KO conferred a lower fasting blood glucose compared with wild type and NOX2 deficient mice. However, STZ treatment induced hyperglycemia similarly among the mouse genotypes, which agrees with the study in p47^phox^-NOXO1 double-KO mice [170]. In the former study, targeting p47^phox^ and NOX1 via genetic ablation and siRNA, respectively, restored endothelium-dependent dilation; therefore, the improvements in vascular function occur independently of improvements in fasting glucose. Nevertheless, the reduction in fasting glucose observed with p47^phox^ KO suggests NOX1 may be detrimental to vascular insulin sensitivity. In a genetically obese model of diabetes (*db^−^/db^−^*) induced by leptin receptor deficiency, diabetic mice demonstrated a worse metabolic profile than lean controls, including elevated glucose, insulin, and triglycerides [172]. Seven days of rosiglitazone treatment were not sufficient to abolish these aberrations, however superoxide production and mRNA levels of NOX1, NOX2, and NOX4 were all reduced. As the loss of endothelial and vascular function precedes the onset of peripheral insulin resistance in high fat diet-induced obesity [180], it can be expected that rapid improvements in vascular oxidative stress occur prior to restorations of circulating glucose and lipids. 

These studies indicate that NOX1 contributes to diminished endothelial function in diabetes via a mechanism that is likely dependent on eNOS uncoupling. However, there are various factors that can lead to dysfunctional eNOS activation (e.g., BH_4_ and L-arginine depletion) and there is a limitation to using p47^phox^ KO as a model of NOX1 KO since NOX2 is also responsive to p47^phox^. Thus, the NOX1-mediated impairments of endothelium-dependent dilation remain unclear. Importantly, the paucity of research regarding the regulation of glucose homeostasis by NOX1 presents numerous opportunities for future investigations such as determining the actions of NOX1 in a post-feeding period.

#### 4.2.2. NOX2

The findings related to the cardiometabolic impairments induced by NOX2 are well-documented in transgenic animal models of diabetes. Excess NOX2-derived superoxide in obesity provokes endothelial dysfunction [175] and exacerbates atherosclerotic lesions in the mouse aorta [173]. In contrast with the NOX1 literature, the metabolic features of NOX2 have been described in dietary obesity and postprandial states [175,176,177,181]. 

The mechanisms of NOX2-mediated oxidative stress and vascular dysfunction were demonstrated in complementary models of endothelial-specific insulin resistance. Sukumar and colleagues [177] generated mice with transgenic overexpression of endothelial-specific mutated (dominant negative) human insulin receptors to exhibit vascular insulin resistance; endothelium-dependent vasodilation was attenuated and superoxide production was significantly elevated. In addition, pulmonary endothelial cells isolated from the transgenic mice, as well as whole aorta and lung, displayed a marked increase in NOX2 mRNA and protein expression, but NOX4 and NOX1 were unaltered. Targeting a siRNA against NOX2 further confirmed that NOX2 was the main source of ROS generation [177]. Selective pharmacological inhibition of NOX2 via gp91ds-tat, however, restored endothelium-dependent dilation, dramatically reduced superoxide production, and improved NO bioavailability. These results were further established by genetic ablation of NOX2 in transgenic mice with endothelial-specific insulin resistance [177]. Intriguingly, targeting of NOX2 via pharmacological inhibition and genetic deletion had no effects on metabolic function: despite restorations in vascular function and mitigation of oxidative stress, body weight, glucose tolerance, and insulin tolerance were all unaffected. This suggests that divergent mechanisms exist between maintenance of glucose homeostasis and the influence of NOX2 on endothelial dysfunction. It can be postulated that NOX2 expressed in other cell types contributes to worsening of insulin sensitivity. Therefore, the investigation of total NOX2 ablation may elucidate interactions between tissues and the ensuing effects on glucose homeostasis.

Studies of global NOX2 KO underscore the contention that NOX2 activity reduces whole-body insulin sensitivity. This notion is supported in NOX2 KO mice that present a dynamic contribution of NOX2 in the development of glucose intolerance [182]. Eight weeks of a high fat diet in NOX2 KO mice attenuated adipose tissue macrophage inflammation and prevented glucose intolerance. However, an additional eight weeks on the same diet conferred a severe metabolic phenotype including excess weight gain, accumulation of liver triglycerides, and insulin resistance [182]. In addition, the onset of insulin resistance upon 16 weeks of a high fat diet was characterized by activated macrophages. Long-term adherence to the high fat diet contributed to a hyper-inflammatory state in adipose tissue and reduced the clearance of dead adipocytes, which led to metabolic dysfunction. The upregulation of NOX2 may, therefore, underlie the macrophage-dependent clearance of dead adipocytes and subsequent protection against insulin resistance. The role of inflammation in insulin resistance is beyond the scope of this review, however immune-based therapies are evolving as effective treatments for T2D [183]. Nevertheless, these results identify that NOX2 can present diverse functions throughout the progression of diet-induced obesity. NOX2 activation may contribute to glucose intolerance during early stages of diet-induced obesity, whereas the upregulation of NOX2 following the onset of insulin resistance may promote clearance of dead adipocytes and reduce ectopic lipid deposition. 

Other studies are in agreement with reports from global NOX2 deletion studies that there is a negative relationship between NOX2 and glucose homeostasis [175,176,181]. Following 16 weeks of a high fat diet, middle-aged mice subjected to NOX2 KO or apocynin treatment were protected from increased superoxide production, MAPK-dependent ERK1/2 activation, reduced insulin receptor expression, and reduced Akt-dependent eNOS phosphorylation in aortas [176]. Perhaps more importantly, NOX2 KO mice were protected from impaired endothelium-dependent dilation, increased fasting blood glucose and glucose intolerance. The protection against insulin resistance is also observed in NOX2 KO mice following a 12-week high fat diet [175]. Therefore, these findings indicate that NOX2-derived superoxide is critical for disruption of vascular insulin signaling, and provokes greater insulin-dependent MAPK versus eNOS activation—a mechanism that is PKC-dependent [184]. The enhanced vasoconstriction effects may explain the attenuations in endothelium-dependent dilation and glucose tolerance observed in wild type mice fed a high fat diet [175,176]; antagonism of the Ang II type 1 receptor restores insulin sensitivity in genetically obese rats [185]. However, it should be mentioned that NOX2 KO mice did not improve hyperinsulinemia induced by a high fat diet [176]. This is an intriguing finding considering the benefits related to fasting glucose and insulin-stimulated glucose disposal, as well as the prevention of insulin receptor downregulation.

Hyperinsulinemia alters the balance between MAPK-dependent vasoconstriction and eNOS-dependent vasodilation in favor of reduced dilation in healthy adults and T2D patients undergoing a hyperinsulinemic euglycemic clamp [186]. Furthermore, Mahmoud et al. [187] demonstrated that physiological concentrations of insulin also increase NOX2-derived superoxide and eNOS uncoupling in isolated skeletal muscle arterioles and human adipose microvascular endothelial cells. The excess generation of ROS observed with high insulin treatment, however, was attenuated by knockdown of NOX2 via siRNA in the cultured cells. Insulin treatment also led to NOX-derived ROS production in myotubes via PKC-dependent translocation of p47^phox^ [188]. These findings indicate that NOX2 is directly susceptible to activation by hyperinsulinemia. However, other animal studies have demonstrated that NOX2 KO either had no effect on [177], or reduced, fasting insulin levels [181]. Considering that T2D presents hyperglycemia and hyperinsulinemia, these findings raise the question whether high concentrations of insulin or glucose in vivo are directly responsible for the detrimental actions of NOX2 in endothelial and metabolic function. The studies in cultured endothelial cells mentioned in Section 4.1 clearly demonstrate that hyperglycemic treatment in the absence of hyperinsulinemia activates NOX2, however the maintenance of glucose homeostasis requires a coordination between various tissues. 

#### 4.2.3. NOX4

NOX4 is the predominantly expressed NOX isoform in endothelial cells [189,190] and primarily generates H_2_O_2_, a hyper-polarizing factor. The distinct localization and ability for NOX4 to generate H_2_O_2_ likely underscore the intriguing ability to promote endothelium-dependent dilation and may explain why NOX4 is generally considered protective of vascular function [114,191]. However, there is contradictory evidence pointing to deleterious functions of NOX4, including eNOS uncoupling via generation of ONOO^−^ in response to Ang II [127]. The dynamic role of NOX4 is similarly exhibited in diabetic settings.

The elucidation of enhanced insulin receptor tyrosine phosphorylation (i.e., activation) by NOX4-derived H_2_O_2_ first implicated the beneficial role of NOX4 in metabolic health [192,193]. A subsequent study by Li and colleagues [194] determined that NOX4 is a crucial regulator of glucose homeostasis. While wild type mice fed a 12-week high fat diet gained significant body weight and abdominal adiposity, NOX4 KO exacerbated the development of obesity within two weeks. This occurred in parallel with significantly greater energy consumption and a marked impairment in energy efficiency (weight gained/calories consumed) compared with wild types. Similarly, NOX4 deficiency led to an accumulation of triglyceride storage in the liver that was matched with a reduced capacity for fatty acid oxidation. In fasting conditions, the liver’s role in glucose homeostasis is to produce glucose from non-carbohydrate sources (e.g., pyruvate) [195], and NOX4 mice displayed impaired gluconeogenesis in response to pyruvate injection. Furthermore, NOX4 KO led to greater elevations in fasting glucose after four weeks of a high fat diet. Interestingly, the hyperglycemia was reversed by week 12, but this was compensated by a robust increase in fasting insulin levels suggesting an induction of insulin resistance. Indeed, results from insulin tolerance tests showed normal insulin sensitivity at week four, and attenuated reductions of glucose following an insulin bolus at week 12. Finally, the adipose tissue of NOX4 KO mice displayed reduced GLUT4 content as well as a pro-inflammatory signature including increased protein expression of NF-κB [194]. Thus, the findings from this study in mice with global NOX4 KO establish a decisive role of NOX4 in mediating metabolic function. Moreover, NOX4 is also considered protective of diabetes-associated atherosclerosis in ApoE KO mice with STZ-induced diabetes [196]. The mechanisms that underlie each of the impairments related to the absence of NOX4 are not all clear. However, considering the vasoprotective effects of NOX4, these findings provide opportunities for future studies to target NOX4 for mitigation of insulin resistance.

Despite evidence of the beneficial features of NOX4, there are contradictory findings demonstrating that NOX4 may be detrimental to insulin sensitivity. NOX4 KO mice fed a chow diet in the Li et al. [194] study displayed an initial reduction in fasting glucose at week four, which increased to levels comparable with wild type mice by week 12. These NOX4 KO mice also gained body weight throughout the intervention, suggesting that NOX4 may convey elevations in blood glucose regardless of adiposity during the initial development of insulin resistance. Enhanced NOX4 mRNA levels and H_2_O_2_ production were also observed in adipose tissue of obese mice with metabolic syndrome, which is in agreement with the suggestion that NOX4 may contribute to insulin resistance [197]. In contrast, adipose-specific deletion of NOX4 prevented insulin resistance and inflammation in mice fed a high fat, high sucrose diet for 16 weeks [198]. Notably, NOX4 activity was initially increased in adipose tissue. However, the adipocytes remained insulin sensitive until long-term adherence to the diet, at which point NOX4 activity declined. This suggests that NOX4 may be upregulated during the onset of insulin resistance as a protective mechanism. Studies in cultured cells have shown that NOX4-derived H_2_O_2_ can increase insulin signaling by oxidation (i.e., inhibition) of protein-tyrosine phosphatase-1B (PTP1B), a major inhibitor of insulin receptor activation (Figure 4) [193]. Whereas in the advanced obese, insulin resistant state, excess nutrient supply to adipose tissue can lead to inflammation and alterations in energy metabolism [5]. Indeed, adipocytes from animals fed the high fat, high sucrose diet displayed a shift toward lipolysis and reliance on fatty acid oxidation, rather than glucose oxidation, despite insulin stimulation [198]. Similarly, reduced adiponectin expression, an anti-inflammatory adipokine, is associated with increased NOX4 activity in T2D patients with CAD [199]. Adiponectin inhibits NOX4 activation via a mechanism that is PI3K/Akt-dependent. This suggests that there is a reciprocal relationship between adiposity and NOX4 activity, and the initial upregulation of NOX4 may be to elicit protective effects related to insulin signaling and vasodilation. 

Thus, these studies in adipose tissue provide evidence of dynamic roles for NOX4 in mediating insulin sensitivity. NOX4 activation during the onset of insulin resistance is likely increased to enhance vascular hyperpolarizing effects and insulin signaling effects; whereas advanced diabetes progression hampers the beneficial effects of NOX4, with oxidative stress instead being induced. The distinct subcellular localization of NOX4 compared with other NOX isoforms may explain the differing functions of NOX4 throughout insulin resistance progression. NOX4 is the only NOX isoform identified with mitochondrial localization [113], and it can be postulated that NOX4-derived H_2_O_2_ in obesity and diabetes promotes excess mitochondrial ROS production, a phenomenon termed ROS-induced ROS release [200]. Overproduction of mitochondrial ROS has been considered an important determinant in the development of T2D, and is increased by hyperglycemia and excess fat intake [201,202]. It is also worth noting that these studies of NOX4 KO and metabolic function did not assess endothelial function and the interplay with eNOS activation. This is an important consideration because NOX4’s beneficial actions in other diabetic complications remain controversial [22].

## 5. Conclusions

In conclusion, it is clear that hyperglycemia and T2D are strong stimulators of NOX1- and NOX2- dependent endothelial dysfunction. Excess superoxide production derived from NOX1 and NOX2 in response to high glucose or a diabetic setting results in impaired endothelial function and insulin resistance. Conversely, NOX4 has emerged as a major intermediary of glucose homeostasis, which is displayed by the array of metabolic derangements that occur when NOX4 is absent. Collectively, the in vitro and in vivo studies have progressed the understanding of mechanisms that activate the different NOX isoforms and the subsequent molecules that are targeted downstream. Uncoupling of eNOS appears to be a shared mechanism between NOX1 and NOX2 that mediates impairments in endothelium-dependent vasodilation and exacerbated oxidative stress. Endothelial NOX2 activation in response to high glucose and insulin alters the balance between Raf/MAPK-dependent vasoconstriction and PI3K/Akt-dependent vasodilation in favor of constriction. Whereas, NOX4 enhances insulin signaling via inhibition of PTP1B to counteract the onset of insulin resistance. The protective effects of NOX4 are further described by the less injurious nature of H_2_O_2_ in the vasculature. However, the interactions between NOX4, NOS/NO and other mediators of endothelial function in T2D remain unknown.

Despite advances in defining the molecular interlinks between diabetic-induced NOX activation and endothelial dysfunction, there remain significant gaps in the literature. The pre-clinical studies have established NOX as central molecules in the regulation of endothelial function; however, it is unclear whether these mechanisms can be translated to humans. Recent developments in methodology to detect NOX-derived ROS in humans by our group provides an opportunity to investigate therapeutic approaches that may reduce endothelial dysfunction in humans [203]. In addition, T2D presents a complicated metabolic environment; therefore, it has been challenging to determine the predominant stimulus of NOX activation. Recent evidence suggests that hyperinsulinemia is the primary cause of NOX-derived superoxide production and endothelial dysfunction, although the relative importance of hyperinsulinemia vs. hyperglycemia have not been determined. Finally, investigations into the interactions between the various NOX isoforms as well as between NOX and other ROS sources will help elucidate the coordination between these sources in specific tissues and subcellular compartments. 

The available evidence indicates the various endothelial NOX isoforms are promising targets to establish the molecular links between CVD and T2D. Physiological NOX activation is a predominant source of ROS production in the endothelium, and helps to maintain insulin signaling and vascular function. However, dysregulated glucose homeostasis has emerged as a major stimulus to provoke vascular pathology. Hyperglycemia and hyperinsulinemia are significant stimuli of aberrant NOX activation and contribute to endothelial dysfunction. The overstimulation of specific isoforms of NOX in settings of T2D creates a cytotoxic environment, where eNOS uncoupling diminishes the ability of the endothelium to promote vasorelaxation and deliver nutrients (Figure 5). Animal models of T2D indicate that antagonism of NOX1 and NOX2 restores endothelium-dependent vasodilation, while NOX4 has dynamic contributions to insulin resistance. Thus, modulations in the magnitude of activation and the relative contribution of each NOX isoform can influence whether there is an advantageous or deleterious effect on vascular function. As the rates of T2D and CVD continue to rise, the evaluation of NOX as mediators of endothelial dysfunction provide important molecular insight to detect early progression of disease. 

## Figures and Tables

**Figure 1 ijms-20-03775-f001:**
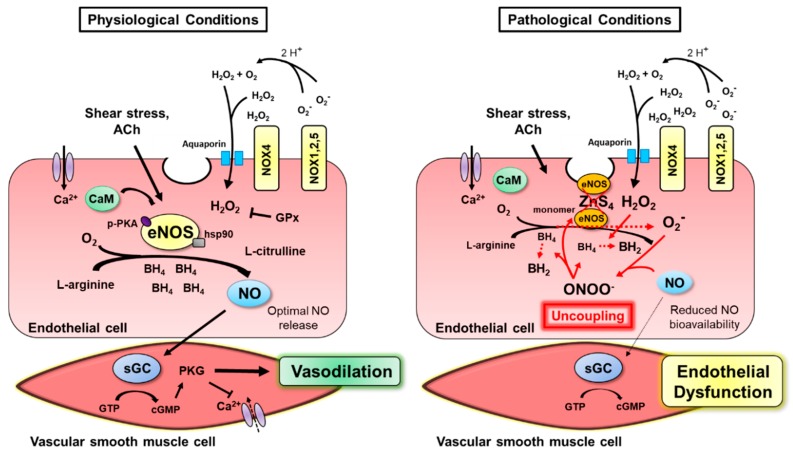
Mechanisms of endothelial nitric oxide synthase (eNOS) uncoupling leading to endothelial dysfunction. (Left panel) Shear stress and acetylcholine (Ach) increase calcium concentrations to mediate calcium-calmodulin (CaM)-dependent and protein kinase A (PKA)-dependent activation of eNOS. PKA phosphorylates eNOS at the Ser-1177/Ser-1179 residue and heat shock protein 90 (hsp90) maintains the eNOS activating conformation as well as releases eNOS from calveolin-1 (Cav-1) at the plasma membrane. L-arginine and molecular oxygen (O_2_) catalyze the activation of eNOS, with the cofactors tetrahydrobiopterin (BH_4_), FMN, FAD, and NADPH (not shown), to produce nitric oxide (NO) and L-citrulline. NO diffuses across the endothelium and targets soluble guanylyl cyclase (sGC) to induce cyclic GMP-dependent activation of protein kinase G (PKG) in vascular smooth muscle cells. The subsequent reduction of intracellular calcium concentrations leads to vasodilation. The NADPH Oxidases 1, 2, and 5 (NOX1, 2, 5) generate superoxide (O_2_^−^), while NOX4 produces hydrogen peroxide (H_2_O_2_) that crosses the plasma membrane and is scavenged by glutathione peroxidase (GPx) (t-bar). (Right panel) However, oxidative stress leads to eNOS uncoupling and endothelial dysfunction. Dysfunctional eNOS activation results in O_2_^−^ production, rather than NO, as shown by the red dashed line. Peroxynitrite (ONOO^−^) is rapidly generated from a reaction between O_2_^−^ and NO, and potentiates eNOS uncoupling. Oxidation of BH_4_ to dihydrobiopterin (BH_2_) (red dashed lines) by ONOO^−^ and H_2_O_2_ limits eNOS substrate availability, and prevents NO production. Further, ONOO^−^ oxidizes the zinc thiolate (ZnS_4_) core of eNOS and disrupts dimerization. Uncoupling of eNOS, therefore, creates a toxic cycle of oxidative stress that reduces NO bioavailability and elicits endothelial dysfunction.

**Figure 2 ijms-20-03775-f002:**
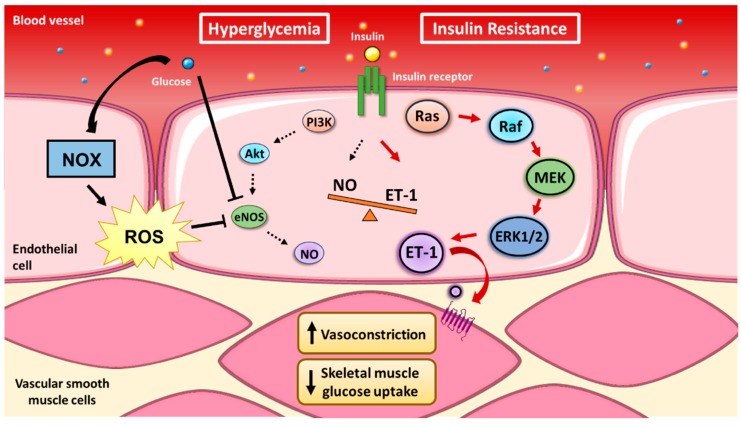
Vascular insulin resistance. There is a reciprocal balance between the divergent branches of endothelial insulin transduction. Insulin stimulates both nitric oxide (NO)-dependent vasodilation and endothelin-1 (ET-1)-dependent vasoconstriction. However, the preferential activation of the Raf/MAPK pathway in vascular insulin resistance leads to excessive vasoconstriction, reduced insulin-stimulated blood flow and reduced insulin-stimulated skeletal muscle glucose disposal.

**Figure 3 ijms-20-03775-f003:**
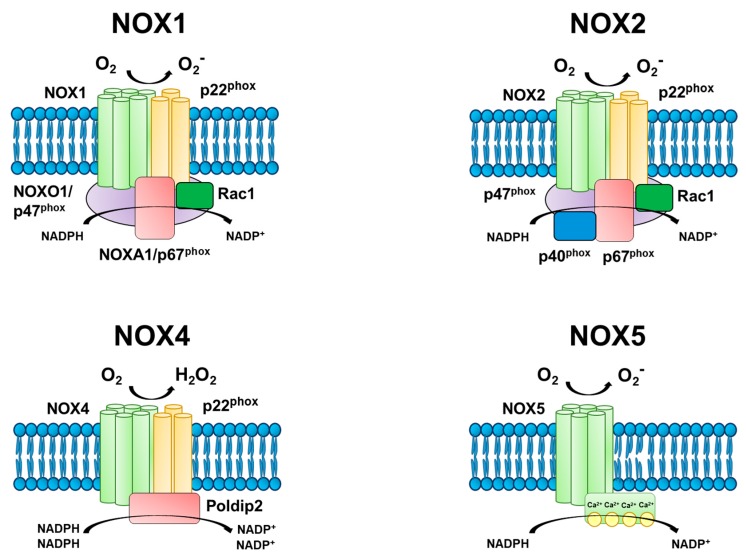
Endothelial NADPH oxidase (NOX) isoforms. The primary function of NOX is the generation of reactive oxygen species (ROS). All four endothelial NOX isoforms are comprised of a catalytic subunit, and NOX1, 2, and 4 require additional subunits for activation. There are several protein-protein interactions involved in producing NOX activity, including interactions between the transmembrane catalytic (i.e., NOX1, NOX2, etc.) and stabilizing (p22phox) subunits with cytosolic organizer (NOXO1/p47phox) and activator (NOXA1/p67phox) subunits. The small GTPase, Rac1, tethers the activator to the plasma membrane, while the organizer acts as a scaffold to maintain interactions between subunits. In contrast, NOX4 is constitutively active in the presence of p22phox and its cytosolic regulator protein polymerase delta-interacting protein 2 (Poldip2). NOX5 does not require additional subunits, and phosphorylation of its cytosolic domain increases the sensitivity to calcium. The electron transfer of cytosolic NADPH to extracellular molecular oxygen generates superoxide (O_2_^−^) in NOX1, 2, and 5, and hydrogen peroxide (H_2_O_2_) in NOX4.

**Figure 4 ijms-20-03775-f004:**
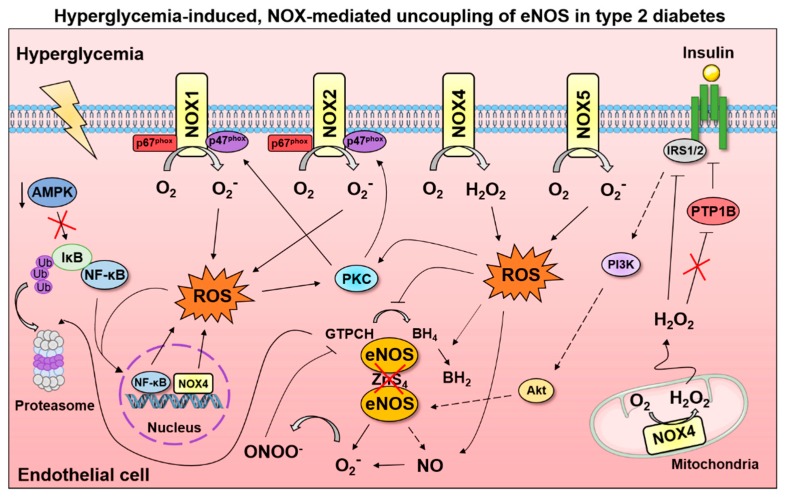
Mechanisms of NOX activation in hyperglycemic conditions. Exposure of cultured endothelial cells to high glucose concentrations elicits activation of NADPH oxidases (NOX). The interactions between specific NOX isoforms and downstream signaling events that lead to endothelial dysfunction are incompletely understood; although several mediators of dysfunctional eNOS activation have been identified. The nuclear factor kappa-light-chain-enhancer of activated B cells (NF-κB) has been reported as a direct activator of NOX4 and a stimulus of NOX1 activation. In contrast, the AMP-activated protein kinase (AMPK) inhibits NF- κB-mediated activation of NOX by preventing proteasomal degradation of the NF-κB inhibitor (IκB). Protein kinase C (PKC) is activated by oxidative stress and phosphorylates the NOX1 and NOX2 subunits (p47^phox^, p67^phox^) to further increase superoxide production. In addition, hyperglycemia-induced oxidative stress depletes the eNOS cofactor, tetrahydrobiopterin (BH_4_), which promotes eNOS uncoupling and enhances superoxide levels. Subsequent peroxynitrite (ONOO^−^) generation oxidizes and promotes proteasomal degradation of the BH_4_ rate-limiting enzyme, guanosine 5′-triphosphate cyclohydrolase I (GTPCH). While hydrogen peroxide (H_2_O_2_) can increase insulin signaling by inhibiting the protein-tyrosine phosphatase-1B (PTP1B), type 2 diabetes may impair the beneficial features of NOX4-mediated H_2_O_2_ production and prevent activation of phosphoinositide 3 kinase (PI3K) by insulin receptor substrate 1/2 (IRS1/2); Akt indicates protein kinase B, BH_2_, dihydrobiopterin.

**Figure 5 ijms-20-03775-f005:**
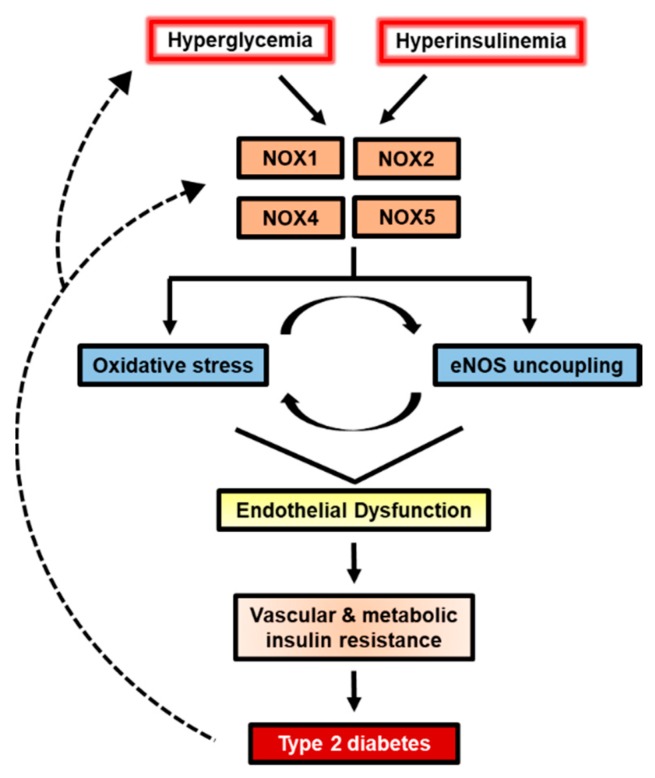
Endothelial dysfunction in type 2 diabetes. Increased NOX activation in settings of type 2 diabetes leads to several vascular and metabolic impairments. Hyperglycemia and hyperinsulinemia are major stimuli of endothelial NOX activation, which contributes to a cytotoxic cycle of oxidative stress and eNOS uncoupling. The altered redox environment subsequently reduces the ability of insulin to perform vasodilatory and glucose transport actions, rendering an essential decrement of endothelial function. Based on human studies, endothelial dysfunction precedes the development of type 2 diabetes, and animal models of type 2 diabetes demonstrate that experimental manipulation of NOX is an effective strategy to mitigate endothelial dysfunction and insulin resistance.

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
