# Peer review of "Endothelial Dysfunction: Is There a Hyperglycemia-Induced Imbalance of NOX and NOS?"

_ijms, 2019, doi:10.3390/ijms20153775_

Reviewer 1 Report

Meza et al. resumed the relation between NOX and NOS in hyperglycemia environment focusing on endothelial cells.

Some issues have been addressed:

1) “Abstract section”: The authors should be added a line as the final “Introduction section” focusing the review description.

2) In the “Introduction section,” the authors have presented all the components that they will address in the following sections. Please, could you add the effect of T2D in chronic kidney diseases (CKD)? CKD is a pathology linked to CVD, T2D, and endothelial dysfunction.

3) In line 82, the authors have presented NOX subunits in the skeletal muscle. What is about these subunits in endothelial cells and damage?

4) In Figure 1, authors have drafted PKG, GTP, and cGMP. Please, could you explain in the text (line 113) too?

4) Please, could the authors add NOX in Figure 1 to clarify that is a source of ROS production?

5) Line 149. Please, could you remove the “knockout” word? Authors have already mentioned in line 132. 

6) In figure 1, there is a mistake in the peroxynitrite abbreviation (right panel). Please, remove “OONO-“ and add “ONOO-.“

7) In figure 1, for better understanding, I suggest adding “Physiological” in the left panel and “Pathology” in the right panel.

8) Could you add in figure 1 RNS?

9) Line 243. Please, could you remove the yellow highlighted?

10) Line 249. About MAPK, could you specify the pathway? In the next paragraph, 2 pathways have been described. Please, could you explain better? PI3K/Akt (PKB) and Ras/Raf/ERK. MAPK is a general name.

11) Please, could you update the figure 2 with the 2 MAPK pathways?

12) Please, could you check superoxide abbreviation along in the review? In the beginning, you have detailed O2- and later on, you have written “superoxide”.

13) In line 320, there is a mistake in “cell´s”.

14) The authors have referred to scavengers among them SOD and catalase (line 351). Please, could add more information about that?

15) Please, could the authors check the liters abbreviation? It is “L”, in capital letter.

16) About the 4.1 section. To clarify better, I suggest a table with the primary data.

17) What is CAPE (line 431)? At least authors should explain that it is a caffeic acid phenethyl ester (CAPE).

18) What is the name of IκBα (line 461)?

19) Please, could you explain the meaning of “…NOX activation yields a milieu…” (line 510)?

20) About section 4.2, for better understanding, the authors should be added a new figure with the essential information from all NOX subunits.

21) Line 528. About diabetic nephropathy, what happens in the kidney with T2D? Please, could you add a paragraph with some data?

22) Line 560. Please use “KO” instead of “knockout”. Please, check all the review.

23) Line 561. Please, change “IFNγ” by “IFN-γ”.

24) Please, check lines 631 and 705 “…by activated macrophages, however. Long-term…” and “…The protective effects of NOX4 are not completely established, however. There is evidence that…”.

25) Line 698. Change by “NF-κB”.

26) About the Conclusions section, the authors should be more concise about the final conclusion of the review. Please, could add a last and resume conclusion? About cite #167, could you remove from Conclusion and add during the review?

27) About Figure 4, could you add in the text? Also, could you add NOX subunits? Please, could you clarify the meaning of T2D, hyperglycemia, and hyperinsulinemia in different boxes?

28) Could you check the reference number 43?

Author Response

Response to Reviewer 1 Comments

1) “Abstract section”: The authors should be added a line as the final “Introduction section” focusing the review description.

Thank you for your suggestion to improve the abstract. Please see the final sentence of the abstract providing the aims of the review.

2) In the “Introduction section,” the authors have presented all the components that they will address in the following sections. Please, could you add the effect of T2D in chronic kidney diseases (CKD)? CKD is a pathology linked to CVD, T2D, and endothelial dysfunction.

Chronic kidney disease is indeed a prominent pathology linked to cardiometabolic disease, and we have added a couple statements to link endothelial and metabolic function to CKD in Line 56.

3) In line 82, the authors have presented NOX subunits in the skeletal muscle. What is about these subunits in endothelial cells and damage?

Thank you for pointing out the need for a statement that addresses the relevance of skeletal muscle NOX expression. Please see Line 83.

4) In Figure 1, authors have drafted PKG, GTP, and cGMP. Please, could you explain in the text (line 113) too?

Please see Line 119 for the revised text, indicating that PKG activation occurs via a cGMP-dependent mechanism.

4) Please, could the authors add NOX in Figure 1 to clarify that is a source of ROS production?

Thank you for your suggestion to add NOX as the source of ROS production. Please see the revised figure.

5) Line 149. Please, could you remove the “knockout” word? Authors have already mentioned in line 132.

Thank you for pointing out this error. The word “knockout” has now been used only at first mention (now on line 169) and is abbreviated to “KO” throughout the rest of the manuscript.

6) In figure 1, there is a mistake in the peroxynitrite abbreviation (right panel). Please, remove “OONO-“ and add “ONOO-.“

Thank you for pointing out this typo in the abbreviation for peroxynitrite. We have made the correction.

7) In figure 1, for better understanding, I suggest adding “Physiological” in the left panel and “Pathology” in the right panel.

Thank you for the suggestion. We have revised Figure 1 to communicate the pathology of eNOS uncoupling more thoroughly.

8) Could you add in figure 1 RNS?

Please see the revisions to Figure 1.

9) Line 243. Please, could you remove the yellow highlighted?

Thank you for pointing out the error. The highlighting has been removed from the text

10) Line 249. About MAPK, could you specify the pathway? In the next paragraph, 2 pathways have been described. Please, could you explain better? PI3K/Akt (PKB) and Ras/Raf/ERK. MAPK is a general name.

Please see Line 279 where we now clarify that the MAPK-dependent signaling involves Ras/Raf/MEK/ERK1/2 activation. Thank you for addressing this.

11) Please, could you update the figure 2 with the 2 MAPK pathways?

Please see the revisions to Figure 2.

12) Please, could you check superoxide abbreviation along in the review? In the beginning, you have detailed O2- and later on, you have written “superoxide”.

Thank you for observing this. We made sure that superoxide is spelled out consistently in the text, and only abbreviated in the figures.

13) In line 320, there is a mistake in “cell´s”.

Please see Line 372. We revised the word “cell’s” to “cellular.”

14) The authors have referred to scavengers among them SOD and catalase (line 351). Please, could add more information about that?

Please see the new paragraph beginning in Line 402. The paragraph describes how NOX4 distinctly generates H2O2 and how this may (or may not) be related to subcellular localization and the presence of SOD. We believe this paragraph adds important evidence to the review. Thus, we thank you for addressing the need to elaborate on this topic.

15) Please, could the authors check the liters abbreviation? It is “L”, in capital letter.

The abbreviation for liters is now capitalized throughout the manuscript.

16) About the 4.1 section. To clarify better, I suggest a table with the primary data.

Thank you for this suggestion. We created a new figure to summarize the findings from the studies reported in section 4.1 (Figure 4). 

17) What is CAPE (line 431)? At least authors should explain that it is a caffeic acid phenethyl ester (CAPE).

Thank you. Please see Line 533. Caffeic acid phenethyl ester (CAPE) is now detailed in the text

18) What is the name of IκBα (line 461)?

Thank you. Please see Line 567. Inhibitor of NF-κB alpha (IκBα) is now detailed in the text.

19) Please, could you explain the meaning of “…NOX activation yields a milieu…” (line 510)?

Thank you for pointing out this error. We added the word “excess” before NOX activation (Line 659) to indicate that it is not necessarily NOX activity leading to deleterious outcomes, but over-stimulation of NOX that is negative.

20) About section 4.2, for better understanding, the authors should be added a new figure with the essential information from all NOX subunits.

Thank you for the suggestion to include a new figure. We believe that with the addition of Figure 4 to the revised manuscript, and the thorough discussion of the studies cited in section 4.2, a new figure would be repetitive. Together, Figures 4 and 5 can provide mechanistic and clinical perspectives of the contributions from NOX in leading to endothelial dysfunction and T2D.

21) Line 528. About diabetic nephropathy, what happens in the kidney with T2D? Please, could you add a paragraph with some data?

Thank you for suggesting the addition of data related to diabetic nephropathy. We agree that this chronic disease is an important and prevalent outcome associated with endothelial dysfunction and type 2 diabetes; however, diabetic nephropathy is out of the scope of the current review. In Section 4.2, we aim to highlight studies that investigate the regulation of glucose homeostasis by endothelial NOX isoforms. As described by Line 678, the roles of NOX and oxidative stress in diabetic vascular complications are important to consider, and have been reviewed elsewhere. We appreciate your feedback on this topic.

22) Line 560. Please use “KO” instead of “knockout”. Please, check all the review.

Thank you for pointing out this error. The word “knockout” has been abbreviated to “KO” throughout the manuscript.

23) Line 561. Please, change “IFNγ” by “IFN-γ”.

Please see Line 711 where we revised the abbreviation to “IFN-γ.”

24) Please, check lines 631 and 705 “…by activated macrophages, however. Long-term…” and “…The protective effects of NOX4 are not completely established, however. There is evidence that…”.

Thank you for addressing these grammatical errors. Please see Lines 779 and 855 for revisions.

25) Line 698. Change by “NF-κB”.

Please see Line 848 for the revision.

26) About the Conclusions section, the authors should be more concise about the final conclusion of the review. Please, could add a last and resume conclusion? About cite #167, could you remove from Conclusion and add during the review?

Please see the final paragraph of the manuscript where we added a more clear and definitive conclusion to our review. We agree that this paragraph improves the review. Thank you for your suggestion.

Citation #167 has been removed from the conclusion and included in Section 4.1(Line 614).

27) About Figure 4, could you add in the text? Also, could you add NOX subunits? Please, could you clarify the meaning of T2D, hyperglycemia, and hyperinsulinemia in different boxes?

Please see the revisions to Figure 4, where the various NOX are included and the figure description clarifies the different boxes.

28) Could you check the reference number 43?

Thank you for finding the mistake with highlighted text

Reviewer 2 Report

This excellent review summarizes the present knowledge on the complex chemical biology of NO and certain other ROS/RNS in endothelial cells and the resulting pathophysiological consequences for vascular smooth muscle cells. Particularly, it lines out how ROS-derived from membrane-bound NOX can trigger the onset of uncoupling of eNOS. ROS derived from NOX seem to cause an initial depletion of BH4, leading to partial uncoupling. The resultant superoxide anions generated by uncoupled eNOS react with NO, leading to the formation of peroxynitrite. This causes further depletion of BH4, leading to complete uncoupling of eNOS and lack of NO release by the endothelial cells, i. e. the pathophysiological state.

I have a few suggestions to further improve this manuscript that already has a very high standard.

1) Some aspects can be formulated more precisely. For example, the authors speak mainly of "ROS" in connection with active NOX, but on the other side, they have very beautifully shown the precise action of different NOX in Figure 3. We know that NOX4 generates H2O2 directly, but that NOX1, 2, 5 generate superoxide anions. These can dismutate to H2O2 spontaneously or through the action of SOD. We also know that superoxide anions cannot pass the cell membrane, but H2O2 can pass the membrane through aquaporins. Therefore, the ROS effect of NOX for the endothelial cell seems to be intruding H2O2.

The authors should mention also, that cells have counteracting antioxidant systems, for example glutathione and glutathione peroxidase (and others)  that partially remove H2O2. If this counteraction is overrun, H2O2 might react with BH4, leading to nonfunctional BH2 .

2) Supoptimal BH4 concentrations cause partial uncoupling of eNOS, i. e. oxygen that during correct function of the enzyme is used to generate NO is converted to superoxide anion.

As these react with NO rapidly, there is a further decrease in NO, and there is formation of peroxynitrite.

Peroxynitrite, according to the literature can also react with BH4, leading to BH2. This effect therefore increases uncoupling. This sequence should be more highlighted.

3) Figure 3: Whereas on molecule of NADPH is sufficient for the generation of one molecule of superoxide by NOX1, 2, 5, the generation of H2O2 by NOX4 should require to molecules of NADPH per molecule of H2O2, together with two protons. Please correct.

4) In addition to some precisions in the text, the right side of Figure 1 should be revised.

Special focus thereby is on the nature and portential action of "ROS", the origin of peroxynitrite and its target, and on the origin of the superoxide anion.

I have summarized some suggestions for possible improvement of Figure 1 in the attachment.

I hope,my suggestions will be useful to optimize Figure 1. For me, it was a great intellectual pleasure to read your mansucript and to work on this presentation.

Author Response

Response to Reviewer 2 Comments

This excellent review summarizes the present knowledge on the complex chemical biology of NO and certain other ROS/RNS in endothelial cells and the resulting pathophysiological consequences for vascular smooth muscle cells. Particularly, it lines out how ROS-derived from membrane-bound NOX can trigger the onset of uncoupling of eNOS. ROS derived from NOX seem to cause an initial depletion of BH4, leading to partial uncoupling. The resultant superoxide anions generated by uncoupled eNOS react with NO, leading to the formation of peroxynitrite. This causes further depletion of BH4, leading to complete uncoupling of eNOS and lack of NO release by the endothelial cells, i. e. the pathophysiological state.

I have a few suggestions to further improve this manuscript that already has a very high standard.

1) Some aspects can be formulated more precisely. For example, the authors speak mainly of "ROS" in connection with active NOX, but on the other side, they have very beautifully shown the precise action of different NOX in Figure 3. We know that NOX4 generates H2O2 directly, but that NOX1, 2, 5 generate superoxide anions. These can dismutate to H2O2 spontaneously or through the action of SOD. We also know that superoxide anions cannot pass the cell membrane, but H2O2 can pass the membrane through aquaporins. Therefore, the ROS effect of NOX for the endothelial cell seems to be intruding H2O2.

The authors should mention also, that cells have counteracting antioxidant systems, for example glutathione and glutathione peroxidase (and others)  that partially remove H2O2. If this counteraction is overrun, H2O2 might react with BH4, leading to nonfunctional BH2 .

Thank you for this insight related to the distinct antioxidant systems that act on superoxide vs. H2O2, as well as the ability for H2O2 to deplete BH4 levels. The revised manuscript includes more detail regarding the differences between ROS and describes the distinct ability of NOX4 to produce H2O2. Please see the two new paragraphs beginning at Line 402.

2) Supoptimal BH4 concentrations cause partial uncoupling of eNOS, i. e. oxygen that during correct function of the enzyme is used to generate NO is converted to superoxide anion.As these react with NO rapidly, there is a further decrease in NO, and there is formation of peroxynitrite.Peroxynitrite, according to the literature can also react with BH4, leading to BH2. This effect therefore increases uncoupling. This sequence should be more highlighted.

Thank you for your suggestion to elaborate on the mechanisms of eNOS uncoupling. Please see the revised paragraph (Line 150) describing that peroxynitrite is a strong stimulus of uncoupling via oxidation of BH4, and that BH4 depletion is a primary determinant of eNOS uncoupling.

3) Figure 3: Whereas on molecule of NADPH is sufficient for the generation of one molecule of superoxide by NOX1, 2, 5, the generation of H2O2 by NOX4 should require to molecules of NADPH per molecule of H2O2, together with two protons. Please correct.

Thank you for this clarification. Please see Figure 3 with the appropriate revision.

4) In addition to some precisions in the text, the right side of Figure 1 should be revised.

Special focus thereby is on the nature and portential action of "ROS", the origin of peroxynitrite and its target, and on the origin of the superoxide anion.

I have summarized some suggestions for possible improvement of Figure 1 in the attachment.

I hope,my suggestions will be useful to optimize Figure 1. For me, it was a great intellectual pleasure to read your mansucript and to work on this presentation.

Thank you for your positive feedback of our review and your time and effort to provide suggestions for improvement of Figure 1.

Reviewer 3 Report

The review paper is original and potentially of interest. However, the quantitative table is lack. Please find the following papers, which could help your job.

[1] “Arterial Waveforms Measured at the Wrist as Indicators of Diabetic Endothelial Dysfunction in the Elderly”, IEEE Trans. on Instrumentation & Measurement, vol.61, no. 1, pp.162-169, 2012. 

[2] “Simultaneous Assessment of Autonomic Nervous and Vascular Endothelial Functions in a Rat Model”, Biomedical Engineering-Biomedizinische Technik, vol.58, no. 2, pp.205-212, 2013.

[3] “Carotenoids in relation to markers of endothelial function and atherosclerosis in young people”, Current Topics in Nutraceutical Research, vol. 11, no. 3, pp.83-90, 2013. 

[4] “Effects of short-term carotenoid-containing multivitamin diet supplementation on vascular health in young adults”, Current Topics in Nutraceutical Research, vol. 14, no. 1, pp. 29-36, 2016. 

I found that this paper is very interesting and that the obtained results are very promising in preventive medicine, however in order to further improve I would only recommend to remove some minor English bugs and to improve more references within the quantitative tables